# Negligible isotopic fractionation of nitrogen within temperate *Zostera* spp. meadows

Douglas G. Russell[1], Wei Wen Wong[1], Perran L.M. Cook[1]

[1]Water Studies Centre, School of Chemistry, Monash University, Clayton, 3800, Australia

*Correspondence to*: Douglas G. Russell (douglas.russell@monash.edu)

**Abstract**

Seagrass meadows form an ecologically important ecosystem in the coastal zone. Excessive nitrogen inputs to the coastal zone pose a key threat to seagrass through eutrophication and associated algal overgrowth. The $^{15}N/^{14}N$ ratio of seagrass is commonly used to assess extent to which sewage derived
nitrogen may be influencing seagrass beds. There have however, been few studies comparing the $^{15}N/^{14}N$ ratios of seagrass beds, their associated sediments and of critical importance, the porewater $NH_4^+$ pool, which is most bioavailable. Here, we undertook a study of the $^{15}N/^{14}N$ ratios of seagrass tissue, sediment porewater $NH_4^+$ pool and the sediment to elucidate the extent of any fractionating processes taking place during organic matter mineralisation and nitrogen assimilation. The study was undertaken within two
coastal embayments known to receive nitrogen from a range of sources including marine, urban and sewage sources. There was extremely close agreement between the bulk sediment $\delta^{15}N$ and seagrass $\delta^{15}N$ ($r^2$ of 0.92 and mean offset of 0.9‰); illustrating a close coupling between the plant and sediment pools. The $\delta^{15}N$ of porewater ammonium was strongly correlated with the $\delta^{15}N$ of both the sediment and seagrass tissue ($r^2$ of 0.89 and 0.85) respectively. The $\delta^{15}N$ porewater $NH_4^+$ minus the $\delta^{15}N$ seagrass tissue ranged
between -1.4 and 7‰ with an average 1.6‰. We suggest the most likely explanation for this was fractionation during assimilation as a consequence of diffusion limitation, although the magnitude of this change was relatively small. Nitrogen fixation may have also contributed a small amount to the observed isotopic depletion of the plants relative to the sediment porewater $NH_4^+$ pool. A consideration of the nitrogen isotope values of the seagrass bed nitrogen pools compared to external sources suggest the
dominant source of nitrogen to seagrass is recycling from within the bed, with a relatively small contribution from water column assimilation, particulate trapping and nitrogen fixation.

# 1 Introduction

Seagrass meadows are widely recognised for their high ecological value, providing a habitat for juvenile fish, stabilising sediment and sequestration of nutrients (Larkum et al., 2006; Nielsen et al., 2004). These ecosystems have been in great decline, due in part to increased nutrient run off and eutrophication (Waycott et al., 2009). Because of the importance of nitrogen in controlling the productivity and eutrophication of coastal environments and seagrass beds, there is great interest in identifying sources of nitrogen to coastal areas.

The ratio of $^{15}N/^{14}N$ (hereafter referred to as $\delta^{15}N$) in seagrass tissue has been widely used to trace nitrogen derived from anthropogenic sources, in particular sewage into seagrass beds (McClelland and Valiela, 1998). The $\delta^{15}N$ of seagrass leaves have also been used as a proxy for nitrogen fixation by seagrass (Hirst et al., 2016; Papadimitriou et al., 2005). A recent global meta-analysis of seagrass $\delta^{15}N$ values found that latitude exerted an overwhelming influence on seagrass $\delta^{15}N$ values, with lighter values being observed in the tropics compared to temperate regions (Christiaen et al., 2014). Possible explanations for this trend included increased nitrogen fixation in tropical waters and an increased predominance of treated sewage as a source of nitrogen in temperate regions. This study highlights the fact that we still have a poor understanding of the factors that control nitrogen isotope ratios in seagrass beds.

A major pool of nitrogen available to seagrasses is ammonium ($NH_4^+$) derived from within the sediment, particularly when water column nitrogen concentrations are low (McGlathery et al., 2001). Seagrasses are able to exploit this pool of bioavailable nitrogen through their ability to assimilate nutrients not just through their leaves, but also their roots. Generally, this pool of $NH_4^+$ within the sediment is present in high concentrations, and as a consequence it is possible that $^{14}N$ will be preferentially assimilated leading to significant isotopic fractionation (Cook et al., 2015). Conversely, it is possible that the distribution of the $NH_4^+$ pool is highly heterogeneous, with low concentrations in the vicinity of roots which would lead to minimal fractionation during assimilation. Previous work by Papadimitriou et al. (2006) in *Zostera noltii* meadows found that the isotopic signature of the seagrass tissue reflected the isotopic signature of porewater $NH_4^+$. This suggests that the benthic pool of nitrogen made a sizeable contribution to the

nitrogen requirements of the seagrass (Papadimitriou et al. 2006). These findings are also consistent with studies in other marine sediments colonised by vegetation such as mangroves. McKee et al. (2002) observed that the fractionation of the residual nitrogen pool in the sediment was dependent on the nitrogen (and nutrient) availability, with limited nitrogen fractionation being observed under nitrogen-limiting

conditions.

In addition to the assimilation of nitrogen leading to potential isotopic fractionation of the nitrogen pool within the sediment, the breakdown (mineralisation) of organic material is another potential source of nitrogen isotope fractionation. Previous studies have found that due to the metabolic discrimination of

$^{14}N$ over $^{15}N$ (e.g. Saino and Hattori, 1987; Altabet and Francois, 1994; Sachs and Repeta, 1999), the residual organic material can end up being isotopically enriched. Previous work by Mobius (2013), Lehmann et al. (2002) and Rooze and Meile (2016) found that in a range of marine environments isotopic fractionation was generally between 2 and 4 ‰.

Nitrogen fixation is also known to be a significant process within seagrass sediments (Welsh, 2000). Previous studies conducted in both Western Port and Port Phillip Bay (Cook et al. 2015; Russell et al. 2016) found that nitrogen fixation was responsible for the production of appreciable amounts of bioavailable nitrogen which was utilised by seagrass to meet a portion of its nitrogen requirements. It has been commonly reported that nitrogen fixation results in little isotopic fractionation (Owens 1988),

therefore in situations where nitrogen fixation is a significant process (such as in seagrass meadows), this can result in an isotopically lighter nitrogen pool. Consequently, one might expect this to lead to a lower $\delta^{15}N$ of $NH_4^+$ within the porewater compared to the sediment.

Given this previous research, it is clear that there are several processes that result in potentially significant

fractionation of isotopes within seagrass meadows, however few studies have explicitly investigated its occurrence or importance. Given the widespread use of $\delta^{15}N$ as a proxy for nitrogen sources and processes within seagrass, it is critical that we understand the extent of nitrogen fractionation within seagrass colonised sediments. To address this, we collected porewater samples for $\delta^{15}N$ analysis of $NH_4^+$, bulk

sediment and seagrass tissues to compare the $\delta^{15}N$ values from a range of seagrass beds influenced by different sources of nitrogen.

## 2 Materials and methods

### 2.1 Study area

A total of 13 sites containing *Zostera muelleri* (except at St. Leonards which contained *Zostera nigricaulis*) were selected for this study, with 10 sites located in Port Philip Bay and 3 sites located in Western Port (Figure 1). Both bays are located in Victoria, Australia and are temperate, intertidal marine embayments. Port Philip Bay is the largest bay in Victoria and has a surface area of ~1930 km$^2$, and Western Port located roughly 55 km south-east of Melbourne and has a surface area of ~650 km$^2$. The sites that were selected from Port Phillip Bay have previously been described in Cook et al. (2015) and exhibit a strong gradient in $\delta^{15}N$ from south to north. Whereas, the sites selected from Western Port have been previously described in Russell et al. (2016) and exhibited a range of nutrient and sediment inputs, as well as differences in areal seagrass coverage. Major sources of nitrogen to Port Phillip Bay include the rivers and drains which contribute ~1000 tonnes per year (TN; total nitrogen) and the Western Treatment plant which contributes ~1000-1500 tonnes TN per year (Harris et al., 1996; Hirst et al., 2016). For Western Port, terrestrial sources of nitrogen from the rivers contribute ~650 tonnes TN per year (Russell et al., 2016).

### 2.2 Sample collection and preservation

Field sampling was carried out in Western Port at intervals of ~2 months between February and November 2016, and sampling in Port Phillip Bay was carried out during August and December in 2016. Differences in the temporal aspect of the sampling regime between Western Port and Port Phillip Bay were a reflection of the logistical difficulties encountered accessing and sampling at the different field sites. The highly tidal nature of the sites selected meant that access to some proved problematic and were only able to be accessed sporadically. Three intact cores containing *Zostera* spp. (65 mm ID × 300 mm long) were

obtained from each site to a sediment depth of ~20 cm. Additionally, intact samples ($n=2-4$) of *Zostera* spp. were obtained from each sample site for elemental (N) and stable isotope analysis ($\delta^{15}N$). All samples were returned to Monash University within 4 hours of sampling.

The overlying water column from the intact cores was removed using a syringe filter leaving only saturated sediment. This sediment was subsequently homogenised and the porewater extracted using of a combination of centrifugation and vacuum filtration. The extracted porewater was subsequently filtered through 0.45 µm and 0.20 µm Sartorius Minisart syringe filters, and frozen until analysis, along with the samples of seagrass. This approach ignores possible depth variation in the porewater $\delta^{15}N$ values,
however, previous work has shown this has minimal variation with depth (Freudenthal et al. 2001; Prokopenko et al. 2006).

### 2.3 Seagrass nitrogen isotope ratios

Seagrass samples were collected from each site, washed by hand with deionised water to ensure that all
detrital and epiphytic material was removed and then dried to a constant weight at 60 °C for 48 hours. The seagrass samples were separated into leaves and roots/rhizomes before being pulverized using a Retsch MM400 ball mill. All analyses were carried out at Monash University on an ANCA GSL2 elemental analyzer interfaced to a Hydra 20-22 continuous-flow isotope ratio mass-spectrometer (IRMS; Sercon Ltd., UK). The stable isotope data was reported in the delta notation ($\delta^{15}N$) and relative to the
isotopic ratio of atmospheric $N_2$ ($R_{Air}$= 0.0036765). The precision of the nitrogen analysis was ±0.2‰ (SD; $n$=5), and ±0.5 µg (SD; $n$=5). To ensure the accuracy of the isotopic results, the following internal standards (ammonium sulphate, sucrose, gelatine and bream) were run concurrently with the seagrass samples. These internal standards have been calibrated against internationally-recognised reference materials (i.e. USGS 40, USGS 41, IAEA N1, USGS 25, USGS 26 and IAEA C-6).

## 2.4 Nutrient analysis ($NH_4^+$, FRP and $NO_X$)

The concentration of $NH_4^+$, filterable reactive phosphorous (FRP) and combined $NO_3^-$ and $NO_2^-$ (hereafter $NO_X$) in the porewater at each site was determined colourimetrically (APHA, 2005) in the National Association of Testing Authorities (NATA) certified laboratory of Monash University (Water Studies Centre). Analysis of ERA-certified reference materials (Lot number P2473-505) indicated the accuracy of the spectrophotometric analysis was within 2% relative error.

## 2.5 Isotopic analysis of porewater $NH_4^+$ ($\delta^{15}N$-$NH_4^+$)

To determine the isotopic signature ($\delta^{15}N$) of the $NH_4^+$ in the porewater, a slightly modified version of the ammonium diffusion method described by Brooks et al. (1989) was used. Incubations were performed in 250 mL Schott laboratory bottles (Schott AG, Mainz, Germany), with target concentrations of ~17.9 to ~28.6 μM N-$NH_4^+$ in a final volume 100 mL. Any required dilutions were carried out using NaCl amended ultra-pure water (~35 ppt.) in order to approximate in situ salinities, and prevent swelling of the membranes housing the acid-traps (Holmes et al., 1998). A subsample of 1 mL was removed from each diffusion bottle prior to the addition of the acid trap in order to determine the actual concentration of N-$NH_4^+$ present in each sample. These samples were filtered through 0.45 μm and 0.20 μm Sartorius Minisart syringe filters and frozen until they were analysed using the indophenol blue method (APHA, 2005). Acid traps were constructed using 4 × 8 mm slices of pre-ashed GF/F paper (Whatman, Buckinghamshire, UK) and acidified with 20 μL of 2.5 M $KHSO_4$, the acidified filter paper was then housed in PTFE membranes (47 mm diameter, 10 μm pore size, Merck Millipore) and crimped shut. These acid traps were added to each diffusion bottle along with ~0.6 g MgO to raise the pH of the solution to ~10. A series of standards were run concurrently using USGS25, USGS26 and IAEA-N1 to ensure that no mass-dependent fractionation effects were encountered. Incubations were carried out at room temperature for 3 weeks on shaker tables at ~135 rpm and the acid traps were then dried in a desiccator in the presence of concentrated HCl for 3 weeks. Afterwards, the dried filter paper was removed from the PTFE membranes and encapsulated in 12 × 8 mm tin capsules (Sercon Ltd., UK). Samples were analysed

for their isotopic signature as well as the total mass of nitrogen using the IRMS described previously. The average recovery obtained for the standards and porewater samples in this study was $100 \pm 5\%$.

**2.5 Statistical analysis**

Two-factor analysis of variance (ANOVA) was undertaken to compare the spatial and temporal differences in porewater nutrient concentrations, as well as seagrass tissue nitrogen isotopic signature. Plots of residuals and boxplots were used to test assumptions of homogeneity and normality of variance (Quinn and Keough, 2002). Where data failed these tests, ln(x) transformation of the data was carried out and then reassessed for homogeneity and normality of variance (all $\delta^{15}N$ values were >1 ‰, and hence

no negative values were obtained with this transformation). In the case of significant responses, post-hoc comparisons were carried out using the TukeyHSD post-hoc test. These analyses were done using R 3.3.0 (R Core Team, 2015). Linear regression analysis was carried out using GraphPad Prism 7 to investigate the relationships between variables. For all analyses, the level of significance required for the rejection of the null hypothesis was set at $p<0.05$.

**3 Results**

**3.1 Nutrient concentrations and isotopic signatures ($\delta^{15}N$) of seagrass and porewater $NH_4^+$**

Porewater concentrations of both FRP and $NO_X$ were consistently low throughout the year in Western Port, with FRP $\leq 2$ µM and $NO_X \leq 27$ µM (Table S1). In contrast, the porewater concentrations of $NH_4^+$

were $1 - 2$ orders of magnitude higher (Figure 2a), with significantly higher concentrations at both Corinella and Rhyll than compared to Coronet Bay ($p<0.05$; Table S2). Furthermore, significant temporal variation in concentrations were also found in Western Port ($p<0.05$; Table S2), with concentrations reaching a maximum during late-autumn/mid-winter. Similarly, in Port Phillip Bay, little FRP and $NO_X$ were detected in the porewater, with the concentrations at all times $\leq 44$ µM and $\leq 27$ µM respectively

(Table S3), whereas the concentration of $NH_4^+$ was up to an order of magnitude higher (Figure 2b). Furthermore, significant spatial variation in $NH_4^+$ concentrations were also evident ($p<0.001$; Table S4)

with the sites in close proximity to terrestrial nitrogen inputs (i.e. Kirk Point and St. Kilda) consistently having the highest porewater $NH_4^+$ concentrations. Significant temporal variation in the porewater concentration of $NH_4^+$ was also evident in Port Phillip Bay ($p$=0.037; Table S4), with the highest concentrations generally found during the winter sampling period.

The $\delta^{15}N$ of seagrass in the context of this study refers to the $\delta^{15}N$ of the roots. There was, however; no significant difference between the $\delta^{15}N$ of the seagrass roots and that of the leaves (Figure S1). Despite the fact that the isotopic signature of nitrogen in seagrass was only found to vary between 2 and 5‰ in Western Port (Figure 2c), significant differences in the isotopic signature were found between the sites

($p$<0.001; Table S5). The heaviest values were consistently found at both Corinella and Rhyll, which were in the closest proximity to human activities and catchments inputs. In contrast, the site at Coronet Bay was the furthest from these inputs and showed a correspondingly low isotopic signature. Statistically significant differences in the isotopic signature of the seagrass were also evident throughout the year ($p$<0.001; Table S5), with the samples obtained during late autumn consistently lower throughout the bay

than at any other time of the year. In contrast, there was appreciable variation in the isotopic signature of the seagrass in Port Phillip Bay (Figure 2d), with values varying from ~2.2‰ to in excess of 16‰ over the course of 2016. The highest isotopic signatures were consistently found in the northerly sites (Kirk Point, Altona and St. Kilda), and south-eastern sites (Blairgowrie and Rosebud), with all isotopic signatures ≥6.9‰. In contrast, the seagrass meadows located in the south-west of Port Phillip Bay (Swan

Bay – North Corio) consistently exhibited the lowest isotopic signatures of between 2.2‰ and 6.4‰. Whilst this resulted in statistically significant differences in isotopic values between sites ($p$<0.001; Table S6), no statistically significant changes in isotopic signatures were observed from winter to summer ($p$=0.350; Table S6).

The isotopic signature of porewater $NH_4^+$ in Western Port was found to exhibit relatively little variation, with values ranging from ~3.9 to 7‰ throughout the course of this study (Figure 2e). Whilst there was no evidence of significant temporal variability in the isotopic signature at each site, there was, however, an apparent north-south gradient in isotopic signatures. In general, the least isotopically enriched

porewater $NH_4^+$ was found in the northern sites (Corinella and Coronet Bay), whilst the highest was found at Rhyll.

Unlike Western Port, appreciable spatial variation in the isotopic signature of porewater $NH_4^+$ was observed throughout Port Phillip Bay. The highest values of between 11.4 and 19.4‰ were consistently observed in the northern sections of the bay from Kirk Point to St. Kilda, whilst sites such as Portarlington and North Corio consistently displayed the lowest values of between 4.2 and 6.4‰ (Figure 2f). The isotopic signature was found to remain reasonably constant throughout the year with the exception of St. Leonards, which displayed an appreciable decrease from winter to summer.

## 3.2 Potential isotopic effects associated with vegetative assimilation and mineralisation

An extremely strong positive and statistically significant linear correlation was observed between the isotopic signatures of seagrass and porewater $NH_4^+$ for all sites throughout this study (Figure 3a; $r^2 = 0.86$, $p<0.001$). The difference between $\delta^{15}N$ porewater $NH_4^+$ and $\delta^{15}N$ seagrass ($\Delta \delta^{15}N_{NH_4^+\ porewater-\ seagrass}$) was statistically significant (paired t-test, $p<0.005$) and ranged between -1.4 to 7‰, with a mean value of 1.6‰. A strong relationship was also observed between the isotopic signatures of sedimentary nitrogen pool and porewater $NH_4^+$ (Figure 3b; $r^2 = 0.89$, $p<0.001$). The difference between $\delta^{15}N$ porewater $NH_4^+$ and $\delta^{15}N$ sediment was statistically significant (paired t-test $p<0.005$) and ranged between -3.6 and 5.9‰, with a mean of 0.9‰. Bulk sediment and seagrass $\delta^{15}N$ values were also tightly correlated with an $r^2$ of 0.92 (Figure 4). There was a statistically significant difference between seagrass $\delta^{15}N$ and sediment $\delta^{15}N$ (paired t-test $p<0.05$) and this ranged between -3.6 to 3.7‰ with a mean value of -0.5‰. There was no relationship between $\Delta(\delta^{15}N_{porewater\ NH_4^+-seagrass})$ and porewater $NH_4^+$ concentration, nor $\Delta(\delta^{15}N_{porewater\ NH_4^+-\ sediment})$ and porewater $NH_4^+$ concentration (Figure 5).

# 4 Discussion

Overall, our study showed very close agreement between the bulk sediment $\delta^{15}N$ and the seagrass tissue $\delta^{15}N$ (Figure 4). This finding is not surprising and consistent with the paradigm that seagrasses rely on sediment derived nitrogen (Barrón et al., 2006), and that a significant fraction of organic matter within seagrass sediments is derived from seagrass itself (Kennedy et al., 2010). The mineralisation of organic matter to $NH_4^+$ and subsequent assimilation by seagrass roots is a critical link coupling the nitrogen $\delta^{15}N$ values in these two pools. To date, there has only been one study on the $^{15}N/^{14}N$ ratios of porewater $NH_4^+$ and its relationship with $^{15}N/^{14}N$ ratios in bulk sediment and vegetation in coastal sediments (Papadimitriou et al., 2006). That study focused on one location in Wales over a seasonal cycle and the present study greatly extends the geographical spread of simultaneous isotope measurements of seagrass tissue, sediment and porewater.

## 4.1 Isotopic signatures of the seagrass pool relative to the porewater $NH_4^+$ pool

Our results showed that on average the seagrass had a $\delta^{15}N$ of 1.6 ‰ less than the $NH_4^+$ in the porewater, with a range of -1.4 to 7‰. Given that there were no significant temporal changes in isotopic signature ($p>0.05$), we assume that the offsets observed here are not an artefact of lags associated with changing isotope pools over time. Three possible explanations for these offsets are considered as follows:

1. Seagrass are assimilating another source of nitrogen through their leaves:

In the following discussion, we assume negligible fractionation of nitrogen during leaf assimilation from the water column. We justify this on the basis that $NH_4^+$ and $NO_3^-$ concentrations in the water column are typically < 1μM at the study sites, and there is therefore unlikely to be significant fractionation taking place.

If we compare the likely $\delta^{15}N$ values of source nitrogen to the seagrass at each of the sites (Table 1), they are typically higher than the porewater values. At the sites in the vicinity of the Western Treatment Plant (Kirk Point and Altona), the $\delta^{15}N$ of the seagrass is ~15‰ which is ~ 8‰ lighter than sewage derived DIN (22.5‰) and therefore direct assimilation of sewage derived by the seagrass leaves seems unlikely

at these sites. The east coast of Port Phillip Bay including sites St Kilda, Rosebud and Blairgowrie are likely to be influenced by the Yarra River plume ($\delta^{15}$N = ~9.7 ‰), Western Treatment Plant and marine sources of nitrogen (Hirst et al., 2016). At all of these sites, the $\delta^{15}$N of seagrass was < 9‰ and therefore assimilation of water column nitrogen is an unlikely explanation for the lower $\delta^{15}$N values observed in seagrass compared to porewater at these sites. The sites in the west of Port Phillip Bay including North and South Corio, Portarlington and Swan Bay are likely to be dominated by marine sources of nitrogen from the water column (Hirst et al., 2016). At all these sites, the seagrass $\delta^{15}$N was < 6.5‰ and therefore assimilation of marine nitrogen from the water column is unlikely to explain the generally lower $\delta^{15}$N in seagrass compared to porewater at these sites. Within Western Port, there was considerable variation in the offset of the seagrass and porewater $\delta^{15}$N values, with ~1/3 of the samples showing a $\delta^{15}$N enrichment in the seagrass compared to the porewater $NH_4^+$. The $\delta^{15}$N values of seagrass within Western Port were always <5‰ and therefore the enriched values of $\delta^{15}$N in seagrass relative to the porewater could be explained by the assimilation of a small amount of terrestrial or marine derived nitrogen from the water column. For the sites where the seagrass was depleted in $\delta^{15}$N relative to the porewater $NH_4^+$, it is unlikely that significant nitrogen assimilation was taking place from the water column. Taken together these results suggest there is very limited direct assimilation of nitrogen from the water column, which is consistent with the low concentrations of inorganic nitrogen in the water column in this region (Russell et al., 2016), and is in agreement with a previous study of seagrass nitrogen sources at an open coastal site (McGlathery et al., 2001).

2. Fractionation of nitrogen during assimilation from the porewater $NH_4^+$ pool:

Handley and Raven (1992) reported that the isotopic fractionation associated with the vegetative assimilation of $NH_4^+$ in a range of environments can vary from 9 to 18‰. Within soils, there is typically a fractionation of only $1 - 2$‰ in association with plant assimilation owing to diffusion limitation (Kendall and McDonnell, 1998; Michener and Lajtha, 2007). At first glance, the notion of diffusion limitation seems at odds with the observation that there were often high concentrations of $NH_4^+$ in the sediment. One possible explanation for this is that $NH_4^+$ concentrations are highly heterogeneous in the sediment resulting in very low concentrations directly within the vicinity of roots where active assimilation is

occurring (Welsh et al., 1997). As such, the assimilation of $NH_4^+$ from the porewater is effectively diffusion limited, leading to minimal isotope fractionation. Evidence to support this comes from the observation that there was no relationship between the $\Delta$ ($\delta^{15}N_{\text{porewater } NH_4^+\text{–seagrass}}$) and the bulk porewater $NH_4^+$ concentration (Figure 5). Such a conclusion is also supported by the lack of an offset between the isotopic signature of the seagrass roots relative to that of the porewater $NH_4^+$ (Figure 3a), with the y-intercept of this graph at ~0.087‰. This finding is in agreement with Papadimitriou et al (2006) who also concluded diffusion limitation was a likely explanation for the small offset observed between porewater and seagrass $\delta^{15}N$ values.

3. Nitrogen fixation within the rhizosphere:

Nitrogen fixation within the rhizosphere of seagrass is well documented and it is thought to be mediated by sulfate reducing bacteria, tightly coupled to the exudation of organic carbon from seagrass roots (Welsh, 2000). As such it is possible that newly fixed nitrogen (which has a $\delta^{15}N$ of ~ 0‰) is rapidly assimilated by seagrass rather than entering the bulk sediment pool. Under this scenario, seagrass nitrogen would become isotopically depleted compared to the porewater $NH_4^+$ pool. To test the plausibility of this, we can undertake some simple calculations with a linear mixing model (Fry, 2006) to estimate the fraction of newly fixed nitrogen assimilated by seagrass assuming a nitrogen fixation end member of 0‰ and using the measured porewater $\delta^{15}N$ values. Using this approach, it was found that nitrogen fixation contributed between 0 and 68% with a mean of 20% to the nitrogen being assimilated by the seagrass in Port Phillip Bay. Direct measurements of nitrogen fixation in Port Phillip Bay have previously suggested nitrogen fixation contributes a maximum of ~15% to nitrogen demand, with a mean of ~5% (Cook et al., 2015). Within Western Port, this mass balance is complicated by the fact that seagrass was sometimes enriched in $\delta^{15}N$ compared to the sediment. In the instances where $\delta^{15}N$ was depleted in the seagrass compared to the porewater, this mass balance yielded a mean contribution of 30%, which is well above previously estimated mean contributions of ~15% by (Russell et al., 2016). We therefore suggest it is unlikely that nitrogen fixation can account for all of the isotopic depletion generally observed within the seagrass relative to the porewater, although we cannot rule out a small contribution.

## 4.2 Isotopic signatures of the sediment pool relative to the porewater $NH_4^+$ pool

In general, the porewater $NH_4^+$ was isotopically enriched compared to the sediment pool by 0.9‰ (Figures 3b and 5b), although this was observed to be highly variable in certain circumstances. Once again this is generally consistent with previous studies of soils (Kendall and McDonnell, 1998; Michener and Lajtha, 2007) and sediment vegetated with seagrass (Papadimitriou et al 2006). Previous work investigating the isotopic fractionation of nitrogen during mineralisation in marine sediments has found that this fractionation is generally in the range of 2 – 4.5‰ (Lehmann et al., 2002; Möbius, 2013 and Rooze and Meile, 2016). This suggests that if mineralisation was having a dominant effect on the $NH_4^+$ isotope pool, then the $\delta^{15}N$ of this pool should be lower than the sediment, which was not the case in this study. Nitrification is another possible process that could lead to an enrichment of the porewater $\delta^{15}N$ pool, however, we believe it is unlikely to explain the fractionation observed here. Nitrification is an obligate aerobic process and it is generally confined to the top few millimetres of sediment owing to the limited penetration of oxygen (Rysgaard et al., 1996). Research has also found that ammonia oxidising bacteria (AOB) are generally outcompeted for available $NH_4^+$ by a range of organisms such macroalgae and bethnic macrophytes (Risgaard-Petersen et al., 2004; Rysgaard et al., 1996). Whilst benthic primary producers such as seagrass can create micro-oxic zones deeper within the sediment (Brodersen et al., 2015; Frederiksen and Glud, 2006), these same seagrasses will also be actively competing with the nitrifiers for bioavailable nitrogen (Vonk et al., 2008). Consistent with this we have measured negligible rates of nitrification coupled to denitrification in intact cores with $^{15}N$-$NH_4^+$ tracer injected into the sediment (Russell et al., 2016). Based on the previous discussion in section 4.1 then, it is most likely that the isotopic enrichment of the porewater compared to the sediment is the result of isotopic fractionation during assimilation by plant roots.

## 5 Conclusions

The strong relationship between the $\delta^{15}N$ values of the seagrass roots, porewater $NH_4^+$ and sediment support the current paradigm that nitrogen is tightly recycled within seagrass beds. On average, nitrogen

within seagrass roots had a $\delta^{15}$N of 1.6 ‰ lower than the porewater, which are most likely explained by isotope fractionation during assimilation of nitrogen from the porewater. This relatively low apparent fractionation factor suggests that seagrass roots are exposed to low sediment $NH_4^+$ concentrations despite high bulk concentrations in the porewater. This apparent discrepancy suggests a high degree of heterogeneity of $NH_4^+$ within the sediment caused by diffusion limitation of nitrogen assimilation. We conclude that although there is a slight offset between seagrass tissue and porewater $NH_4^+$ $\delta^{15}$N values, the $\delta^{15}$N of seagrass is closely coupled to that of bulk sediment $\delta^{15}$N values.

## Acknowledgements

The authors wish to thank Dr. Keryn Roberts for the helpful discussions regarding the application of the ammonium diffusion method used in this study, and to Dr. Adam Kessler, Caitlyn McNaughton and David Brehm for their assistance in the field. This work has been funded by the Australian Research Council (LP130100684), Melbourne Water, Parks Victoria and the Victorian Environmental Protection Authority. Douglas Russell was supported by an Australian Government Research Training Program Scholarship (RTP).

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

**Table 1:** Summary of possible nitrogen isotopic end-members in Western Port and Port Phillip Bay

| Source | Isotopic End-Member (‰) | Reference |
|---|---|---|
| **Port Phillip Bay** | | |
| Nitrogen fixation | 0.0 | Owens, 1988 |
| Oceanic | 6.9 | Russell et al., 2017 |
| Yarra River | 9.7 | Hirst et al., 2016 |
| Western Treatment Plant (WTP) | 22.8 | Nicholson et al., 2011 |
| **Western Port** | | |
| Nitrogen Fixation | 0.0 | Owens, 1988 |
| Oceanic | 6.9 | Russell et al., 2017 |
| Riverine | 9.2 | Russell et al., 2017 |

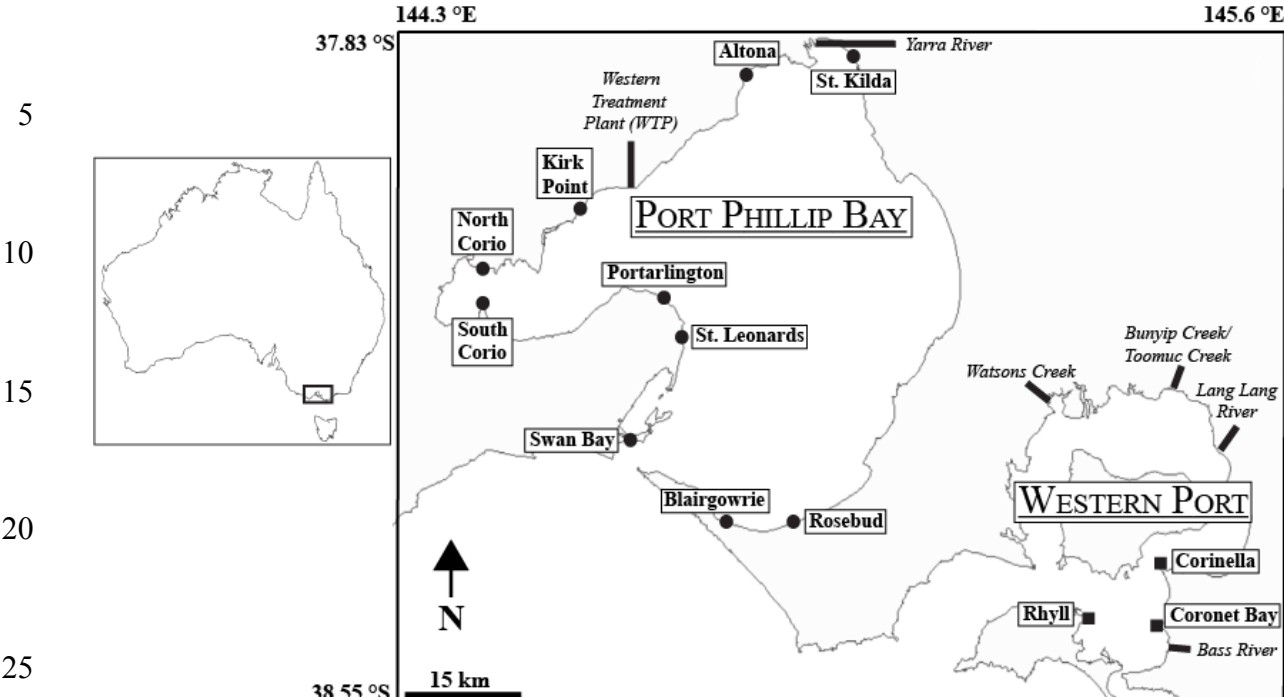

**Figure 1: Western Port and Port Phillip Bay, Australia, showing the field sites. The sites marked with circles were sampled during August and December 2016, and the sites marked with squares were sampled approximately bimonthly over the period February – November 2016.**

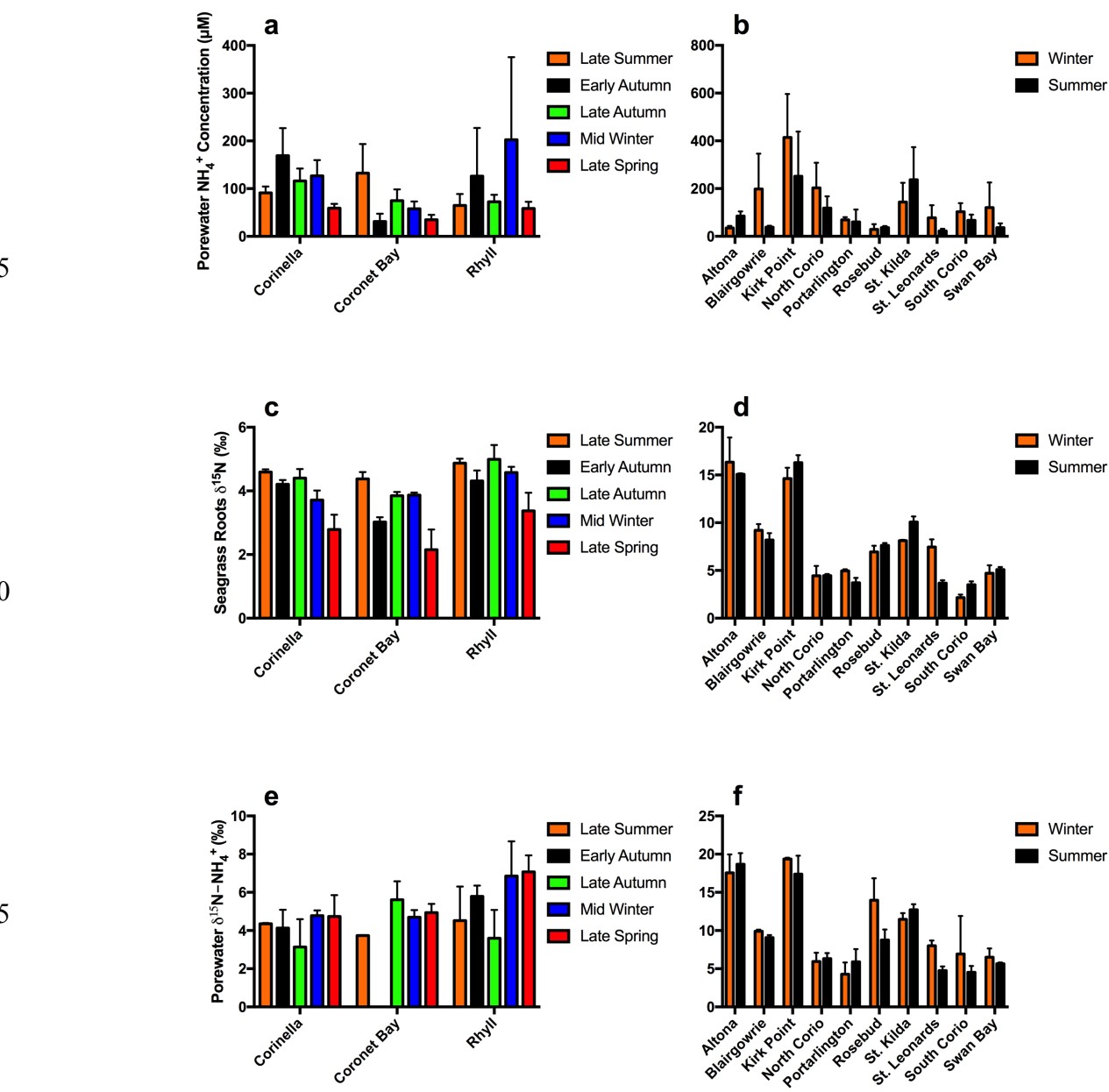

**Figure 2: Porewater NH$_4^+$ concentrations for (a) Western Port and (b) Port Phillip Bay, seagrass root isotopic signature (δ$^{15}$N) for (c) Western Port and (d) Port Phillip Bay, and porewater NH$_4^+$ isotopic signature for (e) Western Port and (f) Port Phillip Bay.** *Note: No results are available for the isotopic signature of porewater NH$_4^+$ in early autumn at Coronet Bay.* **All values are mean ± S.D.**

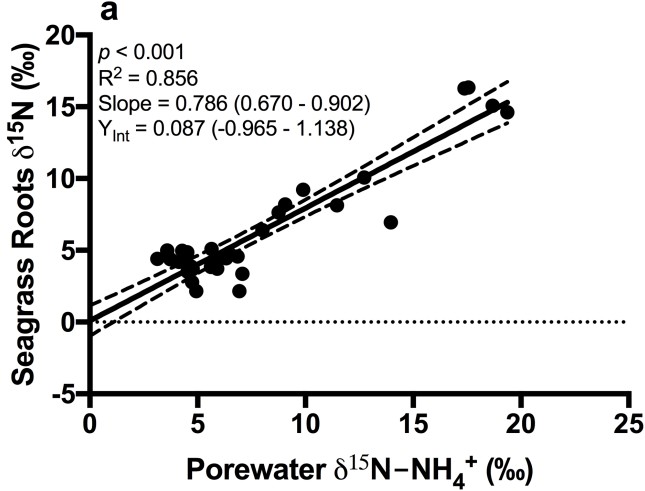

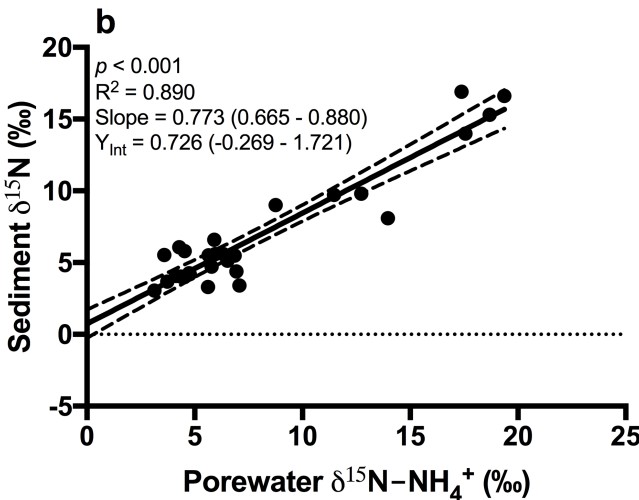

**Figure 3: Plot of the porewater $\delta^{15}$N-NH$_4^+$ against (a) $\delta^{15}$N of seagrass roots and (b) $\delta^{15}$N of the sedimentary nitrogen pool. The 95% confidence intervals of the linear regression are depicted by the dashed lines, and the values in parentheses represent the 95% confidence interval range for the slope and y-intercept.**

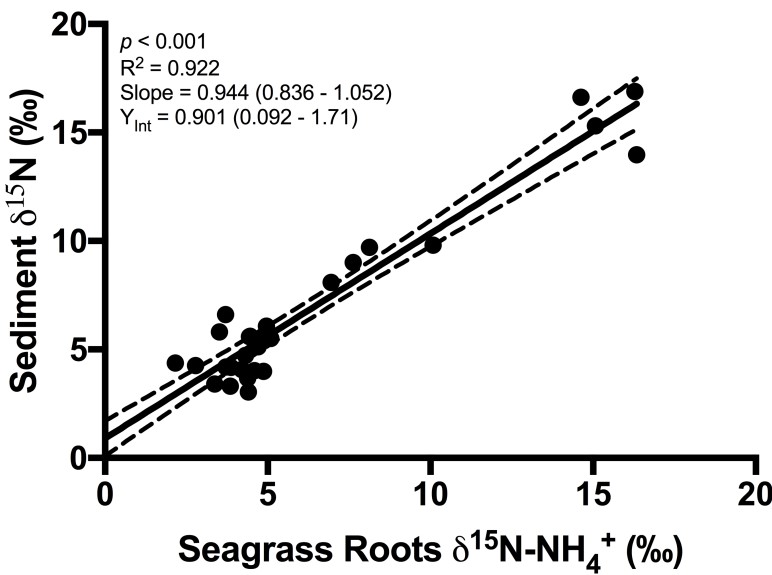

**Figure 4: Plot of seagrass δ¹⁵N against sediment δ¹⁵N.**

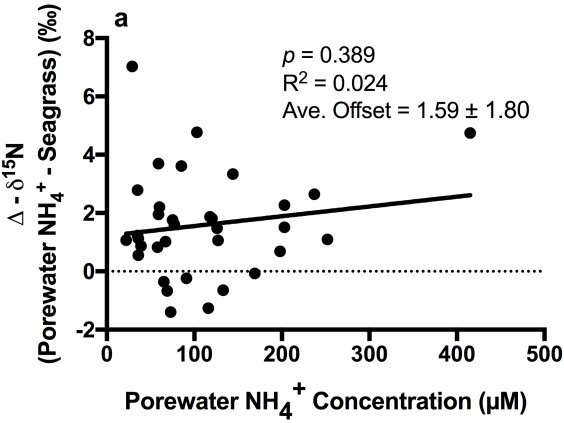

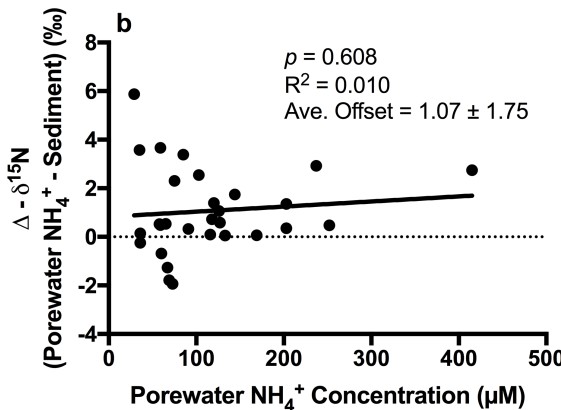

**Figure 5: Plot of porewater NH$_4^+$ concentration against (a) the difference between the porewater δ$^{15}$N-NH$_4^+$ and the seagrass δ$^{15}$N (b) the difference between the porewater δ$^{15}$N-NH$_4^+$ and the bulk sediment δ$^{15}$N.**

