# Peer review of "Negligible isotopic fractionation of nitrogen within temperate *Zostera* spp. meadows"

_Biogeosciences, 2018_

## Referee Comment (RC1) · Anonymous Referee #1 · 29 May 2018

The manuscript by Russell and colleagues reports on a set of field surveys designed to examine patterns in d15N of seagrass, and organic matter and porewater in the sediments underneath seagrass. This is a useful line of inquiry for several reasons, including that seagrasses often suffer from the effects of excess nitrogen, and understanding the mechanisms that generate patterns in d15N of plants will ultimately help us understand broad biogeochemical and ecological patterns. Below I list a set of issues that probably should be resolved before publication, but I suggest that overall the study is a useful contribution.

- It would be good to see some consideration of seagrass physiology and physiognomy in the Introduction. This will help understand why results mostly focused on roots, and also set the context for some of the interpretations and inferences that are consid-

ered in the Discussion. This could include what we know about nitrogen species that seagrasses use, and how they get them.

- Why was the design of the temporal aspect different between the two bays?

- Think more deeply about your hypotheses, and whether the statistical methods used are appropriate. Take the ANOVA: the method is largely inappropriate here, because a two-way fixed factor analysis was used, meaning that the results cannot be broadened beyond the sites and dates surveyed, and the p is largely uninteresting (being overly influenced by sample size). If an ANOVA model is appropriate, I suggest it would be better to use a random-effects model (so that the sites and dates surveyed are considered only a selection of the possible sites and dates that could have been surveyed), and use variance components to examine the important of spatial and temporal variation. Don't forget the interaction term, which is largely ignored here. Also, give the MS in your tables, or the reader doesn't have the information needed to fully examine the results if they wish.

- Also, think about the regression and the paired t-tests. I think the regressions are good, but I also think you can get deeper insights by looking at slope and intercept values, not just r2. From the figures, it seems that confidence intervals around the slop do not overlap 1 – so there isn't a 1:1 relationship, which is very interesting (and probably invalidates the use of paired t-tests).

- Also, log-transformation on data <1 will yield negative results, which probably isn't what you want, did you check that the transformed data make sense?

- Describe what you mean by "sediment solid phase".

- In the methods, more information is needed about the sample collections and preservation, and the seagrass analysis — give the information needed to allow others to repeat the methods, much like you have done for the NH4 analyses. For example, give the sample size (n) for seagrass, describe in more detail how porewater was separated from the water column, how epiphytes were dealt with and how the seagrass was cleaned, what standards were used for the stable isotope analysis.

- Lastly, I would find Figure 2 more useful as a table – think about it.

---

## Referee Comment (RC2) · Anonymous Referee #2 · 11 Jun 2018

Manuscript number: BG 2018-154

"Negligible isotopic fractionation of nitrogen within temperate Zostera spp. meadows."

General comments:

There are several key publications in the area that the authors did not mention in the introduction. Therefore, it is not very convincing and is not giving an overall view to the readers. For example, Papadimitriou et al. (2006) have already reported $\delta$15N in Zostera noltii meadows and $\delta$15N in porewater ammonium with a conclusion that reflected each other. So the "no studies" at line 9, page 1 does not appear justified. Also, the "previous studiES" at line 13, page 1 showing a fractionation of 2‰ of N fractionation during OM mineralization could not be only related to the SINGLE study on

sapropels. The authors are invited to consult Lehmann et al. (2002); Rooze and Meile (2016) where a full description of the N fractionation process during OM mineralization was provided in either marine/lacustrine environments. Therefore, the "uncertainty" mentioned at line 14 is also not justified. These two examples justified the main problems of this manuscript which are the lack of literature documentation leading to the excessive confirmation of confidence (i.e., "no studies" at line 9 page 1, line 17 page 2). The authors are therefore invited to revise the introduction and provide further details on how and why N isotopes are fractionated by geochemical but also biological processes. This lack of a good literature review is also imputable to the quality of the discussion which is not novel and convincing.

Specific comments:

-Section 2.2, 2.3 and elsewhere, the authors are invited to mention the number of samples/replicate collected and number of observation each time a statistical test has been done.

-Which reference materials were used in sections 2.3 to 2.5?

-Page 6 line 11: Again, the authors are invited to revise the "no studies" as it is not quite true.

-The section 4.1 is very hard to follow. The aim of this section is, so far as I understand, is to attribute a reason for the 1.6 ‰ shift in average between seagrass root and porewater. With the approach used and the way the data are shown by the authors, the difficulty is obvious to find a single reason explaining this shift. In fact, the literature shows clearly the vertical gradient of ammonia in porewater, and the bio-irrigation amplifies the heterogeneity of diagenetic reactions. Therefore, a single sample of a broad sediment depth (20 cm) could not be explanatory for the change in N and N isotope for each seagrass root. The correlation in Fig 4 may be enhanced if the authors correct porewater ammonia concentration by the sediment porosity which may give a better idea of the whole N pool, accessible to the plants. However, the section 5.2.2. in Papadimitriou et al. (2006) has very well discussed the N isotope composition in Z. noltii leaves and porewater ammonia. The authors are therefore recommended to shorten and clarify the current section 4.1..

-Similarly, the section 4.2. shows that key publications in the area are missed. Contradictory to what mentioned, there are several work and models on C, N, and their isotopes during mineralization, e.g., (Lehmann et al., 2002; Bouillon et al., 2012; Rooze and Meile, 2016).

-Finally, what is interesting in this study is the correlation between seagrass root, sediment N, and porewater ammonia. Correlations in Fig 3a and 3b show a very similar slope (0.786 vs. 0.773), that means plotting seagrass roots vs. sediment may give a slope of 1. That may lead to a more straightforward conclusion that seagrass roots take the same N isotope signature than sediments rather than trying to explain roots vs. porewater and sediment vs. porewater.

Technical comments:

-Page 5, line 32: The subtraction sign is not a good idea to use in the text as it is confusing with a simple hyphen or a minus, one alternative is using a big delta and having "porewater-seagrass" in subscript.

Reference:

Bouillon S., Connolly R. M. and Gillikin D. P. (2012) Use of Stable Isotopes to Understand Food Webs and Ecosystem Functioning in Estuaries., Elsevier Inc.

Lehmann M., Bernasconi S., Barbieri A. and McKenzie J. (2002) Preservation of organic matter and alteration of its carbon and nitrogen isotope composition during simulated and in situ early sedimentary diagenesis. Geochim. Cosmochim. Acta 66, 3573–3584.

Papadimitriou S., Kennedy H., Rodrigues R. M. N. V., Kennedy D. P. and Heaton T. H. E. (2006) Using variation in the chemical and stable isotopic composition of Zostera

noltii to assess nutrient dynamics in a temperate seagrass meadow. Org. Geochem. 37, 1343–1358.

Rooze J. and Meile C. (2016) The effect of redox conditions and bioirrigation on nitrogen isotope fractionation in marine sediments. Geochim. Cosmochim. Acta 184, 227–239.

---

## Author Comment (AC1) · 23 Jul 2018

REVIEWER #1

We thank the reviewer for their constructive comments:

** It would be good to see some consideration of seagrass physiology and physiognomy in the Introduction. This will help understand why results mostly focused on roots, and also set the context for some of the interpretations and inferences that are considered in the Discussion. This could include what we know about nitrogen species that seagrasses use, and how they get them.

In the revised manuscript the introduction will be re-written to include a more comprehensive discussion of seagrass physiology and physiognomy.

** Why was the design of the temporal aspect different between the two bays?

The different design of the temporal aspect between the different bays was a reflection of the logistical difficulties encountered accessing and sampling at the different sites. The highly tidal nature of the sites selected meant that access to some of the sites proved problematic and were only able to be accessed sporadically – a more detailed description of the sampling design will be provided in the revised manuscript.

** Think more deeply about your hypotheses, and whether the statistical methods used are appropriate. Take the ANOVA: the method is largely inappropriate here, because a two-way fixed factor analysis was used, meaning that the results cannot be broadened beyond the sites and dates surveyed, and the p is largely uninteresting (being overly influenced by sample size). If an ANOVA model is appropriate, I suggest it would be better to use a random-effects model (so that the sites and dates surveyed are considered only a selection of the possible sites and dates that could have been surveyed), and use variance components to examine the important of spatial and temporal variation. Don't forget the interaction term, which is largely ignored here. Also, give the MS in your tables, or the reader doesn't have the information needed to fully examine the results if they wish.

We will re-run our ANOVA ensuring that it takes into account random-effects, the discussion of the model output will also expand upon the results of the interaction term. Currently the outputs of the ANOVA model are presented in the supporting information to the manuscript and these outputs will be updated accordingly.

** Also, think about the regression and the paired t-tests. I think the regressions are good, but I also think you can get deeper insights by looking at slope and intercept values, not just r2. From the figures, it seems that confidence intervals around the slop do not overlap 1 – so there isn't a 1:1 relationship, which is very interesting (and probably invalidates the use of paired t-tests).

We will provide a greater discussion of the slope and intercept values of the figures

presented in the revised version of the manuscript.

** Also, log-transformation on data <1 will yield negative results, which probably isn't what you want, did you check that the transformed data make sense?

Whilst this is true, none of the original $\delta$15N signals were <1, hence log-transforming the data didn't yield negative results.

** Describe what you mean by "sediment solid phase".

We were referring to the sediment upon being dried, we agree that this terminology may be slightly confusing and as a result will substitute "sediment solid phase" with "sediment" in the revised manuscript

** In the methods, more information is needed about the sample collections and preservation, and the seagrass analysis âAËŸT give the information needed to allow others to ËĞ repeat the methods, much like you have done for the NH4 analyses. For example, give the sample size (n) for seagrass, describe in more detail how porewater was separated from the water column, how epiphytes were dealt with and how the seagrass was cleaned, what standards were used for the stable isotope analysis.

The methods section in the revised manuscript will include a more detailed description of the collection, preservation and analysis techniques used relating to the treatment of the seagrass samples.

- Lastly, I would find Figure 2 more useful as a table – think about it.

We will reformat Figure 2 as a table

---

## Author Comment (AC2) · 23 Jul 2018

REVIEWER #2

We thank the reviewer for their constructive comments:

** There are several key publications in the area that the authors did not mention in the introduction. Therefore, it is not very convincing and is not giving an overall view to the readers. For example, Papadimitriou et al. (2006) have already reported $\delta$15N in Zostera noltii meadows and $\delta$15N in porewater ammonium with a conclusion that reflected each other. So the "no studies" at line 9, page 1 does not appear justified. Also, the "previous studiES" at line 13, page 1 showing a fractionation of 2‰ of N fractionation during OM mineralization could not be only related to the SINGLE study on

sapropels. The authors are invited to consult Lehmann et al. (2002); Rooze and Meile (2016) where a full description of the N fractionation process during OM mineralization was provided in either marine/lacustrine environments. Therefore, the "uncertainty" mentioned at line 14 is also not justified. These two examples justified the main problems of this manuscript which are the lack of literature documentation leading to the excessive confirmation of confidence (i.e., "no studies" at line 9 page 1, line 17 page 2). The authors are therefore invited to revise the introduction and provide further details on how and why N isotopes are fractionated by geochemical but also biological processes. This lack of a good literature review is also imputable to the quality of the discussion which is not novel and convincing.

In the revised manuscript both the introduction and discussion will be rewritten to provide the reader with a more comprehensive literature review, with specific references to the manuscript that the reviewer has suggested. We agree that we missed out on one reference that has described nitrogen fractionation processes in seagrass meadows (Papadimitriou et al. 2006). However, as none of the other papers mentioned here deal with nitrogen fractionation specifically in seagrass meadows means that a far greater body of work is required here to understand these fractionation processes. We feel that whilst this study isn't the first to tackle this problem, the lack of overwhelming evidence means that this study is nonetheless an important contribution to the 'limited' body of work that currently exists, and therefore our use of "uncertainty" in this context is justified.

In hindsight we should have reworded the section dealing with the fractionation effect due to mineralisation, however the papers cited here have values between 2.5-4‰ (for Lehmann et al. 2002) and generally 2-3‰ (for Rooze and Meile 2016). Therefore, in the revised manuscript we will revise this statement about mineralisation fractionation to ∼3‰ and add in the references previously mentioned.

** Section 2.2, 2.3 and elsewhere, the authors are invited to mention the number of samples/replicate collected and number of observation each time a statistical test has

been done.

Revised manuscript will have the numbers of samples/replicates and observations each time a statistical test were been done.

** Which reference materials were used in sections 2.3 to 2.5?

Revised manuscript will list all reference materials used

** Page 6 line 11: Again, the authors are invited to revise the "no studies" as it is not quite true.

In the revised version of this manuscript, this statement will be revised

** The section 4.1 is very hard to follow. The aim of this section is, so far as I understand, is to attribute a reason for the 1.6 ‰ shift in average between seagrass root and porewater. With the approach used and the way the data are shown by the authors, the difficulty is obvious to find a single reason explaining this shift. In fact, the literature shows clearly the vertical gradient of ammonia in porewater, and the bio-irrigation amplifies the heterogeneity of diagenetic reactions. Therefore, a single sample of a broad sediment depth (20 cm) could not be explanatory for the change in N and N isotope for each seagrass root. The correlation in Fig 4 may be enhanced if the authors correct porewater ammonia concentration by the sediment porosity which may give a better idea of the whole N pool, accessible to the plants. However, the section 5.2.2. in Pa padimitriou et al. (2006) has very well discussed the N isotope composition in Z. noltii leaves and porewater ammonia. The authors are therefore recommended to shorten and clarify the current section 4.1..

We feel that by using bulk porewater samples over a broad depth give an overall indication of the processes occurring in the sediment, which was the intent of our study; not a fine scale description of fractionation processes. Furthermore, the range of seagrass $\delta$15N was rather narrow (between ∼3-8‰, in comparison our study encompassed a wider range of seagrass $\delta$15N (∼2-16‰. As mentioned by the reviewer, there may be

very localised reactions taking place but the question is whether they significantly contribute to the overall system or are they in effect insignificant on the larger scale? In the revised manuscript we will better articulate our reasoning for looking at the bulk sediment and porewater pools instead of a smaller scale. We will also revise the correlations in Figure 4 as per the reviewer's suggestions to investigate whether stronger correlations are obtained.

** Similarly, the section 4.2. shows that key publications in the area are missed. Contradictory to what mentioned, there are several work and models on C, N, and their isotopes during mineralization, e.g., (Lehmann et al., 2002; Bouillon et al., 2012; Rooze and Meile, 2016). -Finally, what is interesting in this study is the correlation between seagrass root, sediment N, and porewater ammonia. Correlations in Fig 3a and 3b show a very similar slope (0.786 vs. 0.773), that means plotting seagrass roots vs. sediment may give a slope of 1. That may lead to a more straightforward conclusion that seagrass roots take the same N isotope signature than sediments rather than trying to explain roots vs. porewater and sediment vs. porewater.

We agree that we could be clearer about how our study is different to the publications that were listed by the reviewer; in essence, very few studies have used an experimental approach to look at the differences in $\delta$15N between the sediment and porewater NH4+ pools. Section 4.2 will be revised to include a more thorough discussion of the isotopic fractionation effects of mineralisation, with specific references made to the manuscripts that you mentioned. In the revised manuscript, we will investigate the relationship suggested by the reviewer and include a thorough discussion of these results.

** Technical comments: -Page 5, line 32: The subtraction sign is not a good idea to use in the text as it is confusing with a simple hyphen or a minus, one alternative is using a big delta and having "porewater-seagrass" in subscript.

Revised manuscript will be changed to reflect this suggestion

---

## Author Response (AR1)

**Negligible isotopic fractionation of nitrogen within temperate Zostera spp. meadows**

Response to reviewers' comments

We thank the reviewers and the associate editor for their constructive comments. We have addressed the comments by both reviewers (as detailed below) and have revised the manuscript accordingly. Please note that the page and line numbers refer to that in the revised manuscript.

*REVIEWER #1:*

It would be good to see some consideration of seagrass physiology and physiognomy in the Introduction. This will help understand why results mostly focused on roots, and also set the context for some of the interpretations and inferences that are considered in the Discussion. This could include what we know about nitrogen species that seagrasses use, and how they get them.

We have briefly mentioned on how seagrasses acquire nutrients in the introduction section of the revised manuscript.
Page 2 Line 19: 'A major pool of nitrogen available to seagrasses is ammonium ($NH_4^+$) derived from within the sediment, particularly when water column nitrogen concentrations are low (McGlathery et al., 2001). Seagrasses are able to exploit this pool of bioavailable nitrogen through their ability to assimilate nutrients not just through their leaves, but also their roots.'

Why was the design of the temporal aspect different between the two bays?

The different design of the temporal aspect between the different bays was a reflection of the logistical difficulties encountered accessing and sampling at the different sites. The highly tidal nature of the sites selected meant that access to some of the sites proved problematic and were only able to be accessed sporadically. A more detailed description of the sampling design has been provided in the revised manuscript.
Page 4 Line 22: 'Differences in the temporal aspect of the sampling regime between Western Port and Port Phillip Bay were a reflection of the logistical difficulties encountered accessing and sampling at the different field sites. The highly tidal nature of the sites selected meant that access to some proved problematic and were only able to be accessed sporadically.'

Think more deeply about your hypotheses, and whether the statistical methods used are appropriate. Take the ANOVA: the method is largely inappropriate here, because a two-way fixed factor analysis was used, meaning that the results cannot be broadened beyond the sites and dates surveyed, and the p is largely uninteresting (being overly influenced by sample size). If an ANOVA model is appropriate, I suggest it would be better to use a random-effects model (so that the sites and dates surveyed are considered only a selection of the possible sites and dates that could have been surveyed), and use variance components to examine the important of spatial and temporal variation. Don't forget the interaction term, which is largely ignored here. Also, give the MS in your tables, or the reader doesn't have the information needed to fully examine the results if they wish.

We have thought carefully about these comments. The purpose of the analysis undertaken here was to determine whether there was a statistically significant difference between the sites and times sampled as opposed to whether the difference observed here applied more generally across the systems studied. This is consistent with our discussion of the results solely in terms of whether they are different or not.  We do not make any wider regional inferences based on the data. Additionally, the outputs from the statistical analyses requested by the reviewer can be found in the supporting information (Tables S2, S4 – S6).

Also, think about the regression and the paired t-tests. I think the regressions are good, but I also think you can get deeper insights by looking at slope and intercept values, not just r2. From the figures, it seems that confidence intervals around the slope do not overlap 1 – so there isn't a 1:1 relationship, which is very interesting (and probably invalidates the use of paired t-tests).

We have now discussed the slope and intercept values of the figures presented in the revised version of the manuscript.
    Page 12 Line 4: 'Such a conclusion is also supported by the lack of an offset between the isotopic signature of the seagrass roots relative to that of the porewater $NH_4^+$ (Figure 3a), with the y-intercept of this graph at ~0.087‰.'
    Page 14 Line 5: 'We conclude that although there is a slight offset between seagrass tissue and porewater $NH_4^+$ $\delta^{15}N$ values, the $\delta^{15}N$ of seagrass is closely coupled to that of bulk sediment $\delta^{15}N$ values.'

Also, log-transformation on data <1 will yield negative results, which probably isn't what you want, did you check that the transformed data make sense?

Whilst this is true, none of the original $\delta^{15}N$ signals were <1, hence log-transforming the data didn't yield negative results. This has been specifically mentioned in the revised manuscript.
    Page 7 Line 9: '(all $\delta^{15}N$ values were >1 ‰, and hence no negative values were obtained with this transformation)'

Describe what you mean by "sediment solid phase".

We were referring to the sediment upon being dried, we agree that this terminology may be slightly confusing and as a result we have substituted "sediment solid phase" with "sediment" in the revised manuscript.

In the methods, more information is needed about the sample collections and preservation, and the seagrass analysis  T give the information needed to allow others to ˘ repeat the methods, much like you have done for the NH4 analyses. For example, give the sample size (n) for seagrass, describe in more detail how porewater was separated from the water column, how epiphytes were dealt with and how the seagrass was cleaned, what standards were used for the stable isotope analysis.

As suggested by the reviewer, a more detailed description of the collection, preservation and analysis techniques used relating to the treatment of the seagrass samples has been added to the revised manuscript.
    Page 5 Line 5: 'The overlying water column from the intact cores were removed using a syringe filter leaving only saturated sediment. This sediment was subsequently

homogenised and the porewater extracted using of a combination of centrifugation and vacuum filtration.'

        Page 5 Line 14: 'Seagrass samples were collected from each site, washed by hand with deionised water to ensure that all detrital and epiphytic material was removed and then dried to a constant weight at 60 °C for 48 hours.'

        Page 5 Line 21: 'To ensure the accuracy of the isotopic results, the following internal standards (ammonium sulphate, sucrose, gelatine and bream) were run concurrently with the seagrass samples. These internal standards have been calibrated against internationally-recognised reference materials (i.e. USGS 40, USGS 41, IAEA N1, USGS 25, USGS 26 and IAEA C-6).'

        Page 6 Line 5: 'Analysis of ERA-certified reference materials (Lot number P2473-505) indicated the accuracy of the spectrophotometric analysis was within 2% relative error.'

Lastly, I would find Figure 2 more useful as a table – think about it.

We still think that Figure 2 is a better way of presenting our data in order to give the readers an idea on the variabilities of the measured parameters across different sampling sites and seasons. We have however changed the colour scheme of the Figure to make it more presentable and easier to understand.

*REVIEWER #2:*

There are several key publications in the area that the authors did not mention in the introduction. Therefore, it is not very convincing and is not giving an overall view to the readers. For example, Papadimitriou et al. (2006) have already reported $\delta15N$ in Zostera noltii meadows and $\delta15N$ in porewater ammonium with a conclusion that reflected each other. So the "no studies" at line 9, page 1 does not appear justified. Also, the "previous studiES" at line 13, page 1 showing a fractionation of 2‰ of N fractionation during OM mineralization could not be only related to the SINGLE study on sapropels. The authors are invited to consult Lehmann et al. (2002); Rooze and Meile (2016) where a full description of the N fractionation process during OM mineralization was provided in either marine/lacustrine environments. Therefore, the "uncertainty" mentioned at line 14 is also not justified. These two examples justified the main problems of this manuscript which are the lack of literature documentation leading to the excessive confirmation of confidence (i.e., "no studies" at line 9 page 1, line 17 page 2). The authors are therefore invited to revise the introduction and provide further details on how and why N isotopes are fractionated by geochemical but also biological processes. This lack of a good literature review is also imputable to the quality of the discussion which is not novel and convincing.

We have now rewritten the introduction and discussion to provide a more comprehensive literature review, with specific references to the publications that the reviewer has suggested. We agree that we missed out on one reference that has described nitrogen fractionation processes in seagrass meadows (Papadimitriou et al. 2006). However, as none of the other papers mentioned here deal with nitrogen fractionation specifically in seagrass meadows means that a far greater body of work is required here to understand these fractionation processes. We feel that whilst this study isn't the first to tackle this problem, the fact that there has only one previous study at one site on the topic, means this study is nonetheless an important contribution to the 'limited' body of work that currently exists, and therefore our use of "uncertainty" in this context is justified.

: 'Previous work by Papadimitriou et al. (2006) in *Zostera noltii* meadows found that the isotopic signature of the seagrass tissue reflected the isotopic signature of porewater $NH_4^+$. This suggests that the benthic pool of nitrogen made a sizeable contribution to the nitrogen requirements of the seagrass (Papadimitriou et al. 2006). These findings are also consistent with studies in other marine sediments colonised by vegetation such as mangroves. McKee et al. (2002) observed that the fractionation of the residual nitrogen pool in the sediment was dependent on the relative nitrogen (and nutrient) availability, with limited nitrogen fractionation being observed under relatively nitrogen-limiting conditions.'

We have also reworded the section dealing with the fractionation effect due to mineralisation, however the papers cited here have values between 2.5-4‰ (for Lehmann et al. 2002) and generally 2-3‰ (for Rooze and Meile 2016). Therefore, in the revised manuscript we have revised this statement about mineralisation fractionation to ~3‰ and have added in the references previously mentioned.

: 'In addition to the assimilation of nitrogen leading to potential isotopic fractionation of the nitrogen pool within the sediment, the breakdown (mineralisation) of organic material is another potential source of nitrogen isotope fractionation. Previous studies have found that due to the metabolic discrimination of $^{14}N$ over $^{15}N$ (e.g. Saino and Hattori, 1987; Altabet and Francois, 1994; Sachs and Repeta, 1999), the residual organic material can end up being isotopically enriched. Previous work by Mobius (2013), Lehmann et al. (2002) and Rooze and Meile (2016) found that in a range of marine environments isotopic fractionation was generally between 2 and 4 ‰.'

Section 2.2, 2.3 and elsewhere, the authors are invited to mention the number of samples/replicate collected and number of observation each time a statistical test has been done.

We have now added the number of samples/replicates that we used for the statistical test.

: 'Three intact cores containing Zostera spp. (65 mm ID × 300 mm long) were obtained from each site to a sediment depth of ~20 cm. Additionally, intact samples ($n=2 – 4$) of Zostera spp. were obtained from each sample site for elemental (N) and stable isotope analysis ($\delta^{15}N$).'

Which reference materials were used in sections 2.3 to 2.5?

We have now mentioned all the reference materials used for our instrumental analysis in the revised manuscript.

: 'To ensure the accuracy of the isotopic results, the following internal standards (ammonium sulphate, sucrose, gelatine and bream) were run concurrently with the seagrass samples. These internal standards have been calibrated against internationally-recognised reference materials (i.e. USGS 40, USGS 41, IAEA N1, USGS 25, USGS 26 and IAEA C-6).'

: 'Analysis of ERA-certified reference materials (Lot number P2473-505) indicated the accuracy of the spectrophotometric analysis was within 2% relative error.'

Page 6 line 11: Again, the authors are invited to revise the "no studies" as it is not quite true.

In the revised version of this manuscript, this statement has been revised.

The section 4.1 is very hard to follow. The aim of this section is, so far as I understand, is to attribute a reason for the 1.6 ‰ shift in average between seagrass root and porewater. With the approach used and the way the data are shown by the authors, the difficulty is obvious to find a single reason explaining this shift. In fact, the literature shows clearly the vertical gradient of ammonia in porewater, and the bio-irrigation amplifies the heterogeneity of diagenetic reactions. Therefore, a single sample of a broad sediment depth (20 cm) could not be explanatory for the change in N and N isotope for each seagrass root. The correlation in Fig 4 may be enhanced if the authors correct porewater ammonia concentration by the sediment porosity which may give a better idea of the whole N pool, accessible to the plants. However, the section 5.2.2. in Papadimitriou et al. (2006) has very well discussed the N isotope composition in Z. noltii leaves and porewater ammonia. The authors are therefore recommended to shorten and clarify the current section 4.1.

The reviewer has correctly understood the broad aim of this this section. We agree that there are a number of complex factors affecting the possible fractionation of nitrogen and we have broken this down into leaf assimilation from different sources, fractionation during diagenesis and nitrogen fixation. The reviewer has suggested that by sampling the entire sediment pool (rather than profiles) we may have overlooked some important variability of $\delta^{15}N$ within the sediment and have therefore overlooked some potentially important diagenetic processes. The literature does indeed show that ammonium concentrations are highly variable with depth, but this does not necessarily apply to the isotope values, and the limited literature on this has not shown a large variation of $\delta^{15}N$ with depth (Freudenthal et al 2001). We have now added in a new sentence to note we have overlooked depth variability, but we expect this not to be large based on previous measurements.

       Page 5 Line 9: 'This approach ignores possible depth variation in the porewater $\delta^{15}N$ values, however, previous work has shown this has minimal variation with depth (Freudenthal et al. 2001; Prokopenko et al. 2006).'

The reviewer also suggests we shorten this section given that Papadimitriou has also discussed these processes previously. We have carefully read this previous work and we believe the current discussion is warranted because the present study extends to multiple sites, with a much greater variability in the source signature of $\delta^{15}N$. Our discussion also includes a consideration of nitrogen fixation which was not discussed previously.

Finally, the reviewer helpfully suggests we correct the sediment for porosity to tighten up the relationship in Figure 4. Unfortunately, we do not have the data to do this, but in any case, this is only likely to change the data by ~20%. We also do not believe such a correction is warranted because the seagrass are assimilating $NH_4^+$ from the porewater, and therefore it seems most logical to report this concentration rather than per unit of bulk sediment, which the reviewer implies here.

Similarly, the section 4.2. shows that key publications in the area are missed. Contradictory to what mentioned, there are several work and models on C, N, and their isotopes during mineralization, e.g., (Lehmann et al., 2002; Bouillon et al., 2012; Rooze and Meile, 2016). - Finally, what is interesting in this study is the correlation between seagrass root, sediment N, and porewater ammonia. Correlations in Fig 3a and 3b show a very similar slope (0.786 vs. 0.773), that means plotting seagrass roots vs. sediment may give a slope of 1. That may lead to a more straightforward conclusion that seagrass roots take the same N isotope signature than sediments rather than trying to explain roots vs. porewater and sediment vs. porewater.

We agree that we could be clearer about how our study is different to the publications that were listed by the reviewer; in essence very few studies have used an experimental approach to look at the differences in $\delta^{15}N$ between the sediment and porewater $NH_4^+$ pools. Section 4.2 has been revised to include a more thorough discussion of the isotopic fractionation effects of mineralisation, with specific references made to the manuscripts that the reviewer suggested. In the revised manuscript we have also included and discussed the relationship between $\delta^{15}N$ of sediment and $\delta^{15}N$ of seagrass.

      Page 10 Line 2: 'Overall, our study showed very close agreement between the bulk sediment $\delta^{15}N$ and the seagrass tissue $\delta^{15}N$ (Figure 4). This finding is not surprising and consistent with the paradigm that seagrasses rely on sediment derived nitrogen (Barrón et al., 2006), and that a significant fraction of organic matter within seagrass sediments is derived from seagrass itself (Kennedy et al., 2010). The mineralisation of organic matter to $NH_4^+$ and subsequent assimilation by seagrass roots is a critical link coupling the nitrogen $\delta^{15}N$ values in these two pools.  To date, there has only been one study on the $^{15}N/^{14}N$ ratios of porewater $NH_4^+$ and its relationship with $^{15}N/^{14}N$ ratios in bulk sediment and vegetation in coastal sediments (Papadimitriou et al., 2006). That study focused on one location in Wales over a seasonal cycle and the present study greatly extends the geographical spread of simultaneous isotope measurements of seagrass tissue, sediment and porewater.'

Technical comments: -Page 5, line 32: The subtraction sign is not a good idea to use in the text as it is confusing with a simple hyphen or a minus, one alternative is using a big delta and having "porewater-seagrass" in subscript.

The equation has been revised to reflect on this suggestion.

*REFERENCES CITED:*

[revised manuscript text omitted]
 $\delta^{15}N$-$NH_4^+$ and the seagrass $\delta^{15}N$ (b) the difference between the porewater $\delta^{15}N$-$NH_4^+$ and the bulk sediment $\delta^{15}N$.**

---

## Author Response (AR2)

We have carefully considered the reviewers comments and we thank him/her for pushing us on the issue of considering what the slope means. Based on their recommendation we have now focus on this and removed the t-test which they suggest is not appropriate, and we now agree that this hides important information.

In essence, we believe the slope of <1 for the seagrass tissue vs porewater ammonium reflects nitrogen fixation and the slope of < 1 for seagrass for sediment $\delta^{15}N$ versus porewater ammonium reflects isotopically enriched algal material entering the sediment.

We have now modified the discussion, abstract and conclusion to add in this finding, and these changes are visible in track changes.

Regarding the ANOVAs, we believe the statistical significance of temporal and spatial variation in the variables considered here is not important to the focus of the manuscript (and we do not have any central hypothesis in this regard). We now state some of the key general patterns of relevance to the manuscript, most notably the enriched $\delta^{15}N$ values and high porewater $NH_4^+$ concentrations in the vicinity of the sewage treatment plant.

[revised manuscript text omitted]